# Kidney Allograft Function Is a Confounder of Urine Metabolite Profiles in Kidney Allograft Recipients

**DOI:** 10.3390/metabo11080533

**Published:** 2021-08-11

**Authors:** Karsten Suhre, Darshana M. Dadhania, John Richard Lee, Thangamani Muthukumar, Qiuying Chen, Steven S. Gross, Manikkam Suthanthiran

**Affiliations:** 1Bioinformatics Core, Weill Cornell Medicine-Qatar, Doha 122104, Qatar; 2Department of Physiology and Biophysics, Weill Cornell Medicine, New York, NY 10065, USA; 3Division of Nephrology and Hypertension, Department of Medicine, New York Presbyterian Hospital-Weill Cornell Medicine, New York, NY 10065, USA; dmd2001@med.cornell.edu (D.M.D.); jrl2002@med.cornell.edu (J.R.L.); mut9002@med.cornell.edu (T.M.); 4Department of Transplantation Medicine, New York Presbyterian Hospital-Weill Cornell Medicine, New York, NY 10065, USA; 5Department of Pharmacology, Weill Cornell Medicine, New York, NY 10065, USA; qic2005@med.cornell.edu (Q.C.); ssgross@med.cornell.edu (S.S.G.)

**Keywords:** metabolomics, kidney transplantation, allograft rejection, urine analytics, association study, eGFR

## Abstract

Noninvasive biomarkers of kidney allograft status can help minimize the need for standard of care kidney allograft biopsies. Metabolites that are measured in the urine may inform about kidney function and health status, and potentially identify rejection events. To test these hypotheses, we conducted a metabolomics study of biopsy-matched urine cell-free supernatants from kidney allograft recipients who were diagnosed with two major types of acute rejections and no-rejection controls. Non-targeted metabolomics data for 674 metabolites and 577 unidentified molecules, for 192 biopsy-matched urine samples, were analyzed. Univariate and multivariate analyses identified metabolite signatures for kidney allograft rejection. The replicability of a previously developed urine metabolite signature was examined. Our study showed that metabolite profiles can serve as biomarkers for discriminating rejection biopsies from biopsies without rejection features, but also revealed a role of estimated Glomerular Filtration Rate (eGFR) as a major confounder of the metabolite signal.

## 1. Introduction

Kidney transplantation is the treatment of choice for patients who are diagnosed with irreversible kidney failure. However, the full benefits of kidney transplantation are undermined by both immune factors and non-immune complications. Among the immune factors, allograft rejection is the foremost complication, and is a major contributor to allograft failure and the death of the transplant recipient. Among the non-immune factors, infections and malignancy are life-threatening complications. With the introduction of potent immunosuppressive regimens, viral infections have emerged as a major threat, with infection with the cytomegalovirus (CMV) and BK virus being the two leading viral infections in the post-transplant period. Effective anti-viral therapy for treating CMV, but not for BK virus, is currently available. Thus, BK virus-associated nephropathy (BKVN) is an important cause of kidney allograft dysfunction and failure. 

The kidney allograft biopsy is the current diagnostic tool for identifying kidney allograft dysfunction. This invasive procedure has become safer over the years, and the interpretation of biopsy findings has been standardized by almost yearly updates of the Banff classification schema. Nevertheless, bleeding, and even death, are documented complications of kidney allograft biopsy, and biopsy interpretation still suffers from its semi-quantitative nature and inter-observer variability, even among experienced pathologists. Importantly, the immune response that is directed at the allograft is dynamic, and repeat biopsies to capture the kinetics of the anti-allograft repertoire are neither practical nor safe.

The kidney is a major excretory organ, and the glomerular filtration rate (GFR) is either measured or estimated to evaluate kidney function. Multiple factors, including water-soluble drugs and metabolites, are filtered and excreted by the kidneys, and the dosages of drugs are adjusted based on the GFR. Indeed, plasma metabolite concentration may be a more sensitive indicator of the GFR than the clinically used serum creatinine [1].

Noninvasive assessment of kidney allograft status is a major goal in kidney transplantation. Our laboratory pioneered urinary cell mRNA profiling, and demonstrated that the urinary cell levels of mRNAs encoding immunoregulatory proteins and mRNAs encoding cytotoxic proteins are biomarkers of acute cellular rejection (ACR). In the earlier investigation, and in the current one, ACR or acute T-cell-mediated rejection are used as interchangeable terms. Our single-center studies led to a multicenter clinical trials of transplantation (CTOT)-04 study of 485 prospectively enrolled kidney allograft recipients [2]. In the CTOT-04 study, we discovered and validated a urinary cell three-gene signature, consisting of CD3E mRNA, IP-10 mRNA, and 18S rRNA (CTOT-04 signature), which is diagnostic and prognostic of ACR. In a multimodal interrogation of the CTOT-04 study cohort, we identified that a composite signature of the CTOT-04 three-gene signature, and ratios of 3-sialyllactose to xanthosine (3-SL/X) and quinolinate to X-16397, outperform either the mRNA signature or the metabolite signature in diagnosing ACR [3]. 

Metabolite profiles have also been associated with kidney allograft status by others. Ho et al. [4] analyzed samples from adult kidney transplant recipients, using a targeted metabolomics platform, and demonstrated that urinary metabolites distinguished patients with no rejection, sub-clinical rejection, or clinical rejection from each other. Sigdel et al. [5] applied gas chromatography–mass spectrometry (GC-MS) to analyze biopsy-matched urine samples from a pediatric cohort of kidney allograft recipients, and generated metabolite panels to detect graft injury phenotypes. 

In this investigation, we examined whether the metabolites that we previously identified as being associated with ACR are also associated with ACR when urine cell-free supernatants from an independent cohort of kidney allograft recipients are profiled with a new metabolomics platform that is capable of identifying a boarder spectrum of metabolites than in our earlier study (1276 metabolites vs. 749 metabolites). We also investigated whether the metabolite profiles that are associated with ACR are also associated with the other major type of acute rejection, which is the antibody-mediated rejection (AMR). In view of the clinical significance of polyomavirus-associated nephropathy (PVAN), it might of be of interest to explore the metabolite profile of PVAN. While we include all the available samples from this new study in the statistical analysis, to increase the statistical power, we shall address the detailed biomarker analysis of PVAN in a future paper, as its biology is quite different.

In our earlier investigation, we observed a weak association between the metabolite signature and eGFR, but did not have access to the required phenotype data to further investigate this link [3]. In this investigation, we explored this relationship further, including whether eGFR confounds the association between the metabolites and acute rejection. First, we computed linear models for each metabolite with all the available and potentially relevant covariates, including, in particular, eGFR and the CTOT-04 mRNA signature. Then, we computed a range of relevant contrasts, which were defined as differences between the combined estimated means of the selected rejection types, and we have provided the full statistics as Appendix A, so that everything that we do not cover in this paper can be looked up by the interested reader. To include the largest possible number of samples, we focus our discussion on the difference between acute rejection samples, which are a combination of ACR, AMR, and mixed rejection, and non-rejection samples, which are a combination of ATI and normal biopsy samples. As there are more analyses that could be conducted with this dataset than what can be covered in a single paper, we also make the metabolomics data freely available for further analysis by other investigators. Finally, we evaluate the potential of a multi-parameter model to predict the presence of acute rejection in human kidney allografts.

## 2. Results

### 2.1. General Features of This Study

This study was conducted at the NewYork-Presbyterian/Weill Cornell Medical Center (NY-WCM) in New York, NY, USA. The study participants were transplant patients, who all received their kidney transplants at NYP-WCM, and underwent either clinically indicated (for-cause) or surveillance biopsies of their kidney allografts. Urine samples were collected on the day of, and prior to, the biopsy. After the exclusion of samples with incomplete records, 192 samples from 153 unique kidney recipients were included in the statistical analysis. The kidney allograft biopsies were classified as acute cellular rejection/T-cell-mediated rejection (ACR), acute antibody-mediated rejection (AMR), mixed rejection (the presence of both ACR and AMR), acute tubular injury (ATI), or polyomavirus-associated nephropathy (PVAN) (Table 1, Appendix A). Some of the study participants contributed more than one biopsy sample (up to four), which were generally from different time points, and sometimes represented different biopsy diagnoses. For simplicity, and as there were only few patients that provided multiple samples, all the analyses were conducted on a per-sample basis.

Principal component analysis was conducted to identify potential outliers (none were removed), and to visualize the major contributors to the variation in the metabolomics data (Figure 1, Appendix A). The difference in eGFR was a major contributor to the variation in the metabolome. In general, the different rejection types could not be separated from each other on the PCA plot. As the normal/no rejection biopsies were from patients with normal eGFR values, they clustered together, but also overlapped with the rejection samples that had similar eGFR values. Figure 2 shows box-and-whisker plots of the eGFR stratified by biopsy diagnosis. As expected, the biopsies showing ACR, AMR, mixed rejection, ATI, or PVAN, exhibited graft dysfunction, and the eGFR values were less than the eGFR values that were observed in the patients with normal biopsies.

### 2.2. Metabolite Associations with Co-Variates

Linear models, with normalized metabolite levels as the dependent variable, were fitted using limma (see Methods). The independent variables were the biopsy reading (diagnosis as categorial variable), patient age, sex, ethnicity, eGFR, and donor status, including whether they are living or deceased, the type of induction (ATG vs. other) and maintenance immunosuppressive therapy (steroid sparing vs. standard), diabetes (yes/no), and the CTOT-4 urinary cell three-gene signature, as described in [2]. Note that limma provides estimates for each of the individual biopsy readings, and that it then allows us to aggregate these estimates into so-called contrasts, in order to compute p-values between the groups of categorial variables. To increase the statistical power, and to remain consistent with our previous work, we decided to focus our analysis on the contrast between the samples with acute rejection (ACR + AMR + mixed) vs. no-rejection samples (normal + ATI). The contrasts and full summary statistics for pairwise rejection-type comparisons are provided in Appendix A.

Figure 3 shows volcano plots for the associations of all the metabolites with selected co-variates. Biologically expected associations, such as epiandrosterone and androsterone with sex, or glucose with diabetes state, confirm the integrity of the dataset. Several unknown (X-nnnnn) metabolites are associated with the immunosuppressive drug therapy, and are therefore likely to be biochemically related to these drugs. This observation may, in the future, be used to reveal their identity and help extend the metabolite knowledgebase of the Metabolon platform.

The strongest associations were with eGFR measured at the time of the biopsy (Figure 3c). Multiple metabolites from the guanidino and guanine pathways, including 1-methylguanidine and guanidinosuccinate, displayed an inverse correlation with eGFR, whereas 4-guanidinobutanoate, 7-methylguanine, guanidinoacetate, and guanine, showed a positive correlation with eGFR (Figure 4). These observations are in agreement with previous reports that found guanidinosuccinate and methylguanidine to be substantially increased in the plasma and erythrocytes of patients with kidney dysfunction [6]. Other strong associations of eGFR were found with several unknowns, and with retinol. The list of significant associations also included blood metabolites that have previously been reported to correlate with kidney function, such as C-glycosyltryptophan (later identified as C-mannosyltryptophan), N-acetylalanine, and N-acetylcarnosine [1,7]. As kidney function is impaired in most instances of allograft rejection, confounding with eGFR comes as no surprise, and the associations that are reported here might partially be interpreted as coming from a metabolomics study of eGFR at the extreme end of dysfunctional kidneys.

### 2.3. Replication of Earlier Metabolite Associations with ACR

Next, we asked whether the previously identified associations between specific metabolites and ACR biopsy diagnosis are reproducible when they are investigated using an independent cohort of kidney allograft recipients with different types of acute rejection and a new metabolomics platform, with an expanded number of identifiable analytes [3]. Because eGFR was found to be strongly associated with metabolite profiles, and may confound the relationship between the metabolites and acute rejection diagnosis, we performed additional analyses to test the associations, by including eGFR as a covariate to the prediction model. Almost all the associations (22 out of 24) that we previously reported (Table 2) were concordant in the current study, with respect to their directionality, that is, the metabolite being higher or lower in the patients with acute rejection compared to those without also being higher or lower in this replication study. Eleven of the 24 associations were significant at a nominal level (*p*-value < 0.05). Four associations were considered to be fully replicated at a Bonferroni level of significance (*p*-value < 0.05/24), conservatively accounting for multiple testing of the 24 associations. However, only two of the associations, quinolinate and the ratio of quinolinate with the unnamed metabolite X—16397, remained nominally significant after including eGFR into the model. We therefore conclude that, while the previously reported signature appears to remain valid in discriminating between rejection and non-rejection samples, part of the signal might be driven by the general decline in kidney function in the cases of allograft rejection.

### 2.4. Metabolite Associations with Kidney Allograft Rejection (Univariate Analysis)

Our primary aim was to identify urinary metabolites that can noninvasively discriminate kidney allograft rejection from non-rejection samples. To increase the power, statistical estimates for the samples with biopsy diagnosis of ACR, AMR, and mixed ACR/AMR, were grouped together (called acute rejection from hereon) and compared to the samples from patients with normal biopsies, and from the patients with ATI (called no-rejection from hereon). While PVAN samples were not directly used in the calculation of the contrasts, these data points were included in the limma linear model fit, and thereby contributed to the estimation of the overall variance that was attributed to the other covariates. We found that the association signals between the acute- and no-rejection samples were generally weaker than what we observed in our previous study. As discussed above, this is mainly attributed to the confounding of eGFR with kidney rejection. However, this also implies that the metabolites that remained significant after accounting for covariates are likely to be biologically independent from the process that relates to kidney function, as measured through glomerular filtration. The strongest association was found with choline (*p* = 7 × 10^−6^), which is significant at a false discovery rate <1%. The nominal associations (*p* < 0.05) included metabolites such as choline phosphate, acisoga (N-Acetylisoputreanine-gamma-lactam), N1-Methyl-2-pyridone-5-carboxamide, inosine, benzoylcarnitine, quinolinate, phosphoethanolamine, and several unnamed molecules (Figure 5).

### 2.5. Biomarker Potential (Multivariate Analysis)

As the individual metabolite associations with kidney allograft rejection were modest overall, we then asked whether a combination of markers would be able to discriminate acute-rejection from no-rejection samples. For this purpose, we pre-selected the 25 most discriminating metabolites, using a random forest approach, and then further refined the variable set using stability selection [8] (see Methods). The most predictive model had an area under the curve (AUC) of 91.8% (95% CI: 87.6–95.9%), and consisted of nine metabolites, most of which were also individually strongly associated with allograft rejection (Figure 6). Of note is that we had added eGFR and the CTOT-04 mRNA signature to the set of potentially discriminating variables, which allowed them to be selected by the method, but neither was selected in the prediction model. Interestingly, the stress marker cortisone was included in the multivariate marker selection, although its individual association with allograft rejection was not significant.

## 3. Discussion

In this investigation, we asked whether metabolites that were measured in biopsy-matched cell-free urine supernatants distinguish kidney allograft recipient patients with acute-rejection biopsies from patients with biopsies without histological features of acute rejection. We found that kidney function, as measured by eGFR, is an important confounding factor of metabolite profiles. Sekula et al. investigated the blood-based markers of kidney function, using a non-targeted metabolomics platform [1]. Many of the metabolites that they reported in that study were also found here, in urine. The metabolites that were most strongly associated with eGFR were from the guanidino and guanine pathways. Our examination of the replicability of our previously identified associations, from the multi-center CTOT-04 study, identified quinolinate as a metabolite that remains significant, even after accounting for all the available sources of variation, including ethnicity, age, sex, diabetes, and medication used for the induction and maintenance of immunosuppression and, importantly, eGFR. While the association of the ratio quinolinate/X—16397 also replicated, it should be noted that the *p*-gain, that is, the increase in the strength of association when using ratios [9], does not.

The validated increase in quinolinate in patients with acute rejection biopsies is of interest, since this metabolite has been considered a “double-edge” sword. Quinolinate is an intermediate metabolite in the tryptophan metabolism contributing to the de novo biosynthesis of nicotinamide adenine dinucleotide (NAD^+^), which is pivotal to energy and critical cellular processes [10]. Importantly, de novo NAD^+^ biosynthesis impairment has been linked to acute kidney injury in humans [11]. Thus, higher quinolinate levels may reflect impaired NAD^+^ biosynthesis during an episode of acute rejection, and restoration of NAD^+^ synthesis may be of benefit in this setting. It is also possible that the accumulation of quinolinate, which is considered a toxic metabolite, may be a contributory factor to graft dysfunction during an episode of acute rejection.

The current study was not designed to identify mechanisms for the alterations in metabolite concentration, such as the increase in quinolinate. However, in our ongoing RNA sequencing studies of human kidney allografts undergoing acute rejection, we identified that the mRNA for quinolinate phosphoribosyltransferase (QPRT) is significantly reduced in acute rejection biopsies compared to biopsies without acute rejection changes. The reduced expression of QPRT in the rejecting allograft is a biologically plausible mechanism for increased quinolinate, since QPRT is central to NAD+ synthesis from quinolinate.

This current study differs from our previous study, in considering both ACR and AMR rejection instead of ACR alone, and using an updated version of the Metabolon (Durham, NC) platform that includes metabolites that have not been investigated in the context of kidney rejection before. We could therefore identify new associations that had not been observed before, and assure that these metabolite profiles were most likely not the result of impaired kidney function alone. Using a multivariate analysis with variable selection, we affirm that the metabolites that were included in the prediction model discriminate acute- from no-rejection samples, with an AUC of 91.8%.

Our investigation has several strengths. The biospecimens were collected from kidney transplant patients who all received their kidney transplants at a single center and were managed with standardized immunosuppressive protocols, thereby minimizing the variabilities that are intrinsic to studies involving a multicenter study cohort. Each cell-free urine supernatant that was analyzed in this study was matched to a kidney allograft biopsy, ensuring “ground truth” about the kidney allograft status, where each biopsy was interpreted by highly experienced kidney pathologists and classified using the Banff classification schema, thereby minimizing the inter-observer variability in biopsy interpretation. The urine cell-free supernatants were stored in a −80 °C biorepository, and were not thawed prior to metabolomic profiling. It is noteworthy that unbiased metabolite profiling was performed, using the highly robust and state-of-the-art Metabolon platform.

Our study also has several limitations. Kidney allograft function was defined using estimated GFR, rather than measured GFR, and the “best” formula to be used to estimate GFR is an area of evolving science. Importantly, the inclusion of race in the eGFR measurement is being vigorously addressed by the professional societies, and in this study, a substantial number of kidney transplant recipients were black recipients and the eGFR was calculated using race as one the variables [12]. Then, there is always the issue of association analysis, that is, whether the metabolite profile is altered by GFR or whether the GFR is altered by metabolites, or both. With respect to the metabolites measured, the measurements are relative in nature, and the translation of any marker signature would benefit from the development of a targeted assay for absolute quantification, followed by a reevaluation of the model coefficients using absolute values. While we included all the covariates that might explain part of the variation, and that were available to us, other non-identified factors are likely to co-exist. The variable selection methods depend on multiple parameters, and may yield differing results. The prediction model we developed here should, therefore, be interpreted as one of many possible ones. In order to allow others to build on our study, and potentially develop additional accurate models for prediction, we have provided the full dataset in Appendix A.

As a correlate, the unknown metabolites that are associated here with the drugs used for the maintenance of immunosuppression, are likely to be metabolites of immunosuppressive drugs. This information could be used in the future, to facilitate the identification of unidentified molecules using a mass-spectrometry method, following the approach suggested by Krumsiek et al. [13].

In summary, our investigation of the association between metabolite profiles and acute rejection in human kidney allografts, in addition to replicating the directionality of the 22 of the 24 metabolites that have previously been associated with ACR [3], identified a candidate signature model consisting of 9 metabolites that discriminated kidney allograft recipients, with the biopsies showing acute rejection from the recipients without acute rejection. The current study also emphasizes the importance of considering GFR in linking metabolite concentrations to kidney allograft biopsy findings. We hypothesize that metabolites, besides serving as biomarkers, may suggest novel therapeutics, such as the use of nicotinamide adenine mononucleotide to promote NAD+ biosynthesis and improve allograft function.

## 4. Materials and Methods

### 4.1. Study Cohort, Biopsy Diagnosis and Biospecimens

The study cohort consisted of 153 unique kidney allograft recipients. All recipients received their kidney allografts at the NewYork-Presbyterian/Weill Cornell Medical Center and were managed with standardized immunotreatment protocols. Appendix A is a summary of baseline characteristics of kidney allograft recipients stratified by biopsy diagnosis. Among the 153 recipients, 99 were males (64.7%) and 54 were females (35.3%). Most recipients were either white (45.1%) or black (30.7%) recipients. The major cause of kidney disease resulting in end-stage kidney disease was diabetes mellitus. Among the recipients, 24 (15.7%) were repeat transplants, and 80 (52.3%) were recipients of deceased donor kidneys and 73 (47.7%) were recipients of living donor kidneys.

A total of 273 urine samples were collected at the time of kidney allograft biopsies and urine cell pellets and cell-free urine supernatants were prepared using a standard protocol as previously reported and stored at −80 °C, as previously reported [2]. Metabolite analysis was performed (see below) with aliquots of supernatants that were never thawed prior to metabolite profiling.

Among the 273 biopsy-matched cell-free urine supernatants, 192 samples were included in data analysis (see below). Appendix A is a summary of biopsy diagnosis and biopsy-associated parameters. The biopsies were classified by highly experienced renal pathologists using the Banff classification scheme and were masked for metabolite data. Among the 192 biopsies, 22 biopsies from 22 recipients were classified as acute T-cell-mediated rejection biopsies (ACR), 16 biopsies from 16 recipients as acute antibody-mediated rejection (AMR), 14 biopsies from 14 recipients as mixed rejection, a combination of both ACR and AMR, 36 biopsies from 32 recipients as polyomavirus-associated nephropathy (PVAN), 51 biopsies from 49 recipients as acute tubular injury (ATI) and 53 biopsies from 29 recipients as normal biopsies. Among the 192 biopsies, 137 were clinically indicated biopsies (for cause biopsies) and the remaining 55 were surveillance biopsies. Time since transplantation to biopsies, serum creatinine at the time of biopsy and Banff histological semiquantitative scores are included in Appendix A. Data regarding the presence or absence of IgG donor HLA-specific antibodies at the time allograft biopsy are also provided there.

This is a single-center cross-sectional study of kidney allograft recipients enrolled at the time of kidney allograft biopsy. The biopsies were performed between August 2005 and December 2014, and biopsy-matched urine cell-free supernatants were stored in our IRB-approved biorepository prior to profiling.

### 4.2. Metabolomics

Then, 273 urine samples were submitted to Metabolon (Durham, NC, USA) for analysis on their non-targeted HD4 platform. The Metabolon platform has been extensively used and described in previous studies [14,15]. Briefly, prior to extraction, recovery standards were added for QC purposes. Proteins were precipitated with methanol. Samples were then split into four aliquots for liquid-phase mass-spectrometry analysis; there were two for analysis by two separate reverse-phase (RP)/UPLC-MS/MS methods with positive-ion mode electrospray ionization (ESI), one for analysis by RP/UPLC-MS/MS with negative-ion mode ESI, and one for analysis by HILIC/UPLC-MS/MS with negative-ion mode ESI. Median relative standard deviation (RSD) of the internal standards was 4% and for pooled patient samples the RSD was 6%.

After excluding samples with incomplete medical records, data for 1251 metabolites measured in 192 samples were retained for statistical analysis. Further, 674 metabolites were annotated using Metabolon’s proprietary methods and 577 were of unknown identity. These samples were from 153 unique patients, whereof 128 provided a single sample and 27 provided between two and four samples. Samples were processed by Metabolon following established protocols and procedures. Raw metabolite values (ion counts) were scaled by run-day, so that run-day medians equal the overall median (OrigScale). The OrigScale values for each sample were then normalized by sample osmolality and scaled to set the median equal to 1. Lastly, missing values were imputed with the minimum of each metabolite (OsmoNormImpData).

### 4.3. Statistical Data Analysis

R (version 4.0.4) and rstudio (version 1.4.1103) were used for statistical data analyses. Except for the CTOT-04 replication, linear models were computed using R package “limma” (version 3.46.0) and “autonomics” (version 0.99.22) with the following model: *Metabolite~Diagnosis + eGFR.at.Biopsy + Recipient_Age + Recipient_Gender + Recipient_Ethnicity + Transplant.Type + Induction + Maintenance.Immunosuppres. + CTOT4.Signature + Primary.Diabetes*, where “*Metabolite*” are the OsmoNormImpData metabolite values that were further log10-transformed and scaled to a mean of zero and a standard deviation of one (z-scored). “*Diagnosis*” is a categorial with factors “ACR”, “AMR (incl chronic)”, “ATI”, “mixed rejection”, “Normal/No Rejection”, and “PVAN (incl follow up)”. “*Recipient_Ethnicity*” has factors “Black” and “Non-Black”, “*Transplant.Type*” has factors “Deceased” and “Living”, “*Induction*” has factors “other” and “Thymo”, and “*Maintenance.Immunosuppres.*” has factors “SS” and “SM”.

For the CTOT-04 replication, to be consistent with the approach taken in the previous study, linear models were computed using the R function “lm” (R stats package). Unscaled and unimputed metabolite levels (OrigScale) were first log-scaled, then z-scored, and then outliers more than 4 standard deviations from the mean were removed. A linear model with metabolite levels as outcome and ACR case/control as independent variable was used, considering “ACR”, “AMR” and “mixed rejection” as cases, and “Normal/No Rejection” and “ATI” as controls.

For the multivariate biomarker selection, R packages “stabs” (version 0.6-4) and “randomForest” (version 4.6-14) were used. Briefly, the 25 most discriminating metabolites from a randomForest analysis (computing 500 trees) were used to fit general linear models with stability selection (stab.glmnet.lasso with parameters q = 12, cutoff = 0.6, family = “binomial” and sampling.type = “MB”). The eGFR and CTOT-04 mRNA signature were added to the independent variable list.

## Figures and Tables

**Figure 1 metabolites-11-00533-f001:**
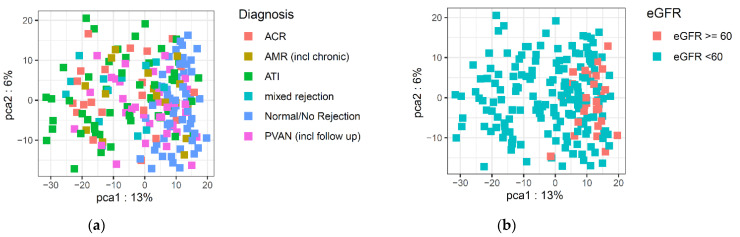
Principal component analysis (PCA) plots of the urinary metabolomics data, colored by “diagnosis” (**a**) and “eGFR” (**b**).

**Figure 2 metabolites-11-00533-f002:**
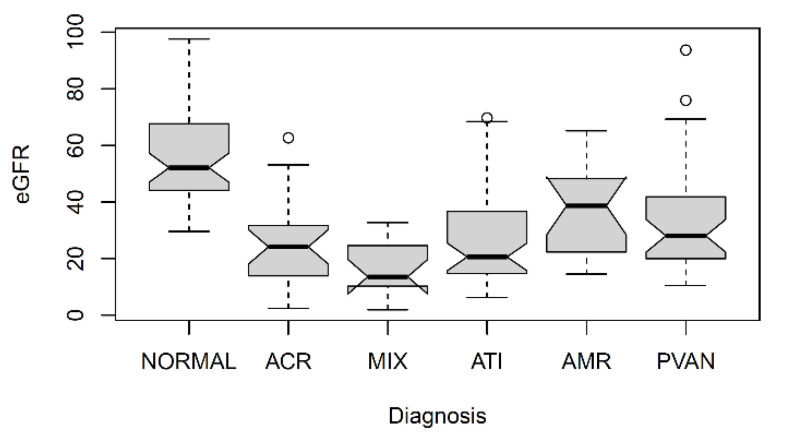
Box-and-whisker plots of eGFR by biopsy diagnosis. Number of samples per box are given in Table 1. Notches indicate the 95% confidence interval of the mean (indicated by the black horizontal line).

**Figure 3 metabolites-11-00533-f003:**
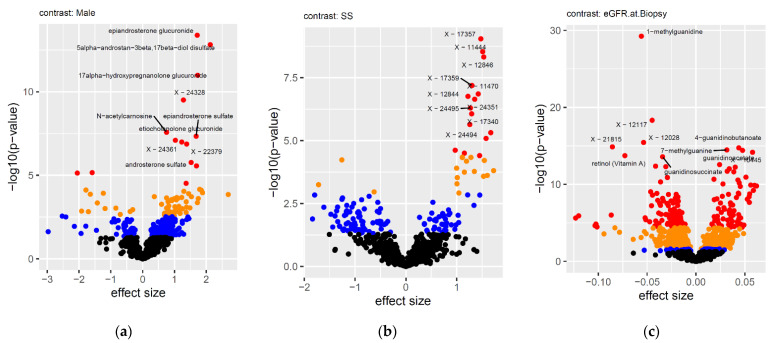
Volcano plots of (**a**) sex, (**b**) maintenance immunosuppressive therapy without steroids (SS) or with steroids, and (**c**) eGFR at biopsy. The significance level of the associations is indicated by color as follows: black = not significant, blue = *p*-value < 0.05, orange = false discovery rate (FDR) < 5%, red = Bonferroni level, *p*-value < 0.05/1276.

**Figure 4 metabolites-11-00533-f004:**
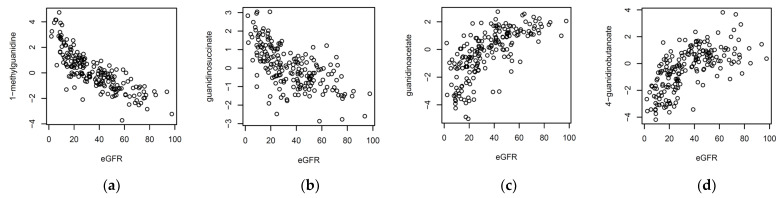
Scatterplots of (**a**) 1-methylguanidine, (**b**) guanidinosuccinate, (**c**) guanidinoacetate, and (**d**) 4-guanidinobutanoate versus eGFR, summary statistics (effect size, standard error, and *p*-value) for the metabolite associations with eGFR are in Appendix A.

**Figure 5 metabolites-11-00533-f005:**
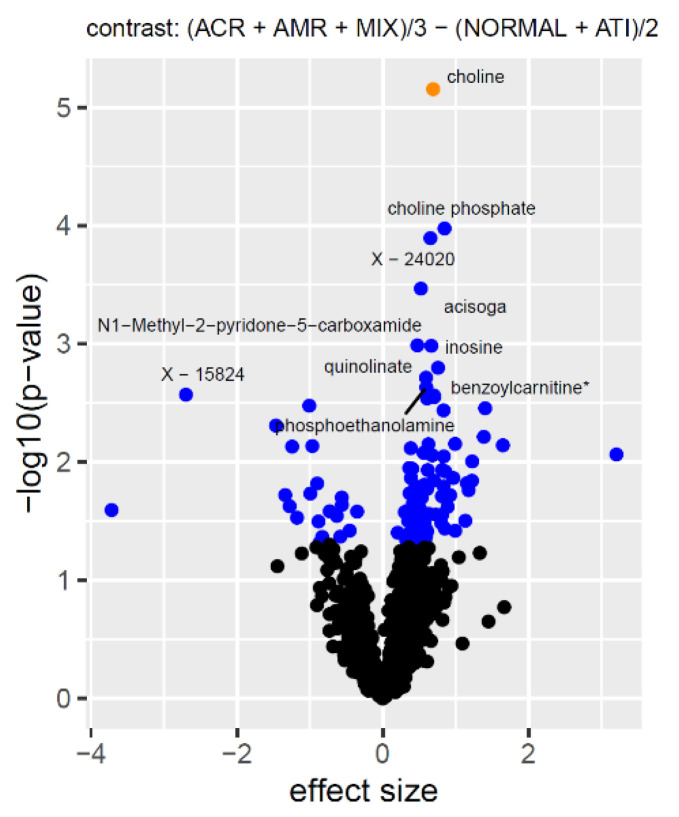
Volcano plot for the difference between estimated means of (ACR + AMR + mixed) and (normal + ATI). Significance levels of the metabolite associations are indicated by colors as described in Figure 3. The ten strongest associations are labeled. Full association data is available as Appendix A.

**Figure 6 metabolites-11-00533-f006:**
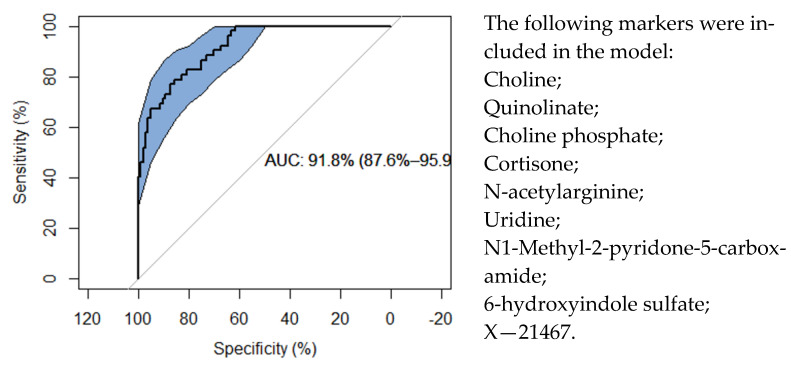
ROC curve for a nine-marker model to discriminate acute rejection from no-rejection samples. We used random forest to pre-select the 25 most discriminating metabolite markers and then constructed a logistic classification model using glm.lasso with stability selection, including eGFR and the CTOT-04 signature in the set of explanatory variables. Note that while proposed as model terms, neither eGFR nor the CTOT-04 signature were selected for inclusion in the prediction model. The area under the curve (AUC) and its confidence interval are provided.

**Table 1 metabolites-11-00533-t001:** Summary of sample characteristics and patient demographics. Additional details are provided in Appendix A.

Kidney Allograft Biopsy Diagnosis	Biopsy (n)
ACR	22
AMR	16
Mixed rejection	14
ATI	51
Normal/no rejection	53
PVAN	36
Total N of samples	192
Demographics of kidney Transplant recipients	
Diabetes: yes/no	44/109
Sex: male/female	99/54
Ethnicity: black/non-black	47/106
Donor: living/dead	73/80
Induction: ATG */other	126/27
Maintenance: SS **/standard	105/48
Total N of unique patients	153

* Anti-thymocyte globulin; ** steroid-sparing maintenance regimen.

**Table 2 metabolites-11-00533-t002:** Replication of associations reported in [3]. Associations displaying identical directionality of the association with acute rejection (same trend), nominal (*p*-value < 0.05) or Bonferroni (*p*-value < 0.05/24) significance, and nominal significance after accounting for eGFR as a co-variate are marked by a cross (“X”).

Label	Beta (N) JASN	*p*-Value ^1^ JASN	Beta (N)	*p*-Value ^2^	*p*-Value ^3^, Incl. eGFR	Same Trend	Nominal	Bonferroni	Incl. eGFR
quinolinate/X—16397	0.89 (248)	7.3 × 10^−9^	0.59 (156)	5.5 × 10^−4^	6.7 × 10^−3^	X	X	X	X
quinolinate/ 4-hydroxymandelate	0.88 (248)	1.1 × 10^−8^	0.44 (125)	0.022	n.s.	X	X		
neopterin/xanthosine	0.9 (234)	2.0 × 10^−8^	0.29 (154)	n.s.	n.s.	X			
3-sialyllactose/xanthosine	0.86 (242)	5.0 × 10^−8^	0.23 (154)	n.s.	n.s.	X			
neopterin/X—16570	0.84 (235)	9.7 × 10^−8^	0.18 (156)	n.s.	n.s.	X			
neopterin/N1-methylguanosine	0.83 (238)	1.2 × 10^−7^	0.41 (154)	0.014	n.s.	X	X		
3-sialyllactose/X—16397	0.77 (247)	5.9 × 10^−7^	−0.10 (156)	n.s.	n.s.				
proline	0.63 (245)	4.9 × 10^−5^	0.44 (156)	0.013	n.s.	X	X		
quinolinate	0.59 (247)	1.8 × 10^−4^	0.76 (156)	1.1 × 10^−5^	3.6 × 10^−3^	X	X	X	X
isoleucine	0.58 (244)	2.1 × 10^−4^	0.19 (155)	n.s.	n.s.	X			
X—13723	−0.57 (235)	3.7 × 10^−4^	−0.31 (156)	n.s.	n.s.	X			
X—12117	0.55 (242)	4.9 × 10^−4^	0.40 (156)	0.014	n.s.	X	X		
1,2,3-benzenetriol sulfate (1)	−0.55 (241)	5.8 × 10^−4^	−0.21 (139)	n.s.	n.s.	X			
leucine	0.52 (246)	9.9 × 10^−4^	0.24 (154)	n.s.	n.s.	X			
pipecolate	0.52 (239)	1.1 × 10^−3^	0.57 (136)	2.2 × 10^−3^	n.s.	X	X		
paraxanthine	−0.63 (158)	1.1 × 10^−3^	−0.61 (119)	1.9 × 10^−3^	n.s.	X	X	X	
1,5-anhydroglucitol (1,5-AG)	0.57 (197)	1.1 × 10^−3^	0.11 (136)	n.s.	n.s.	X			
kynurenate	0.51 (245)	1.3 × 10^−3^	−0.15 (156)	n.s.	n.s.				
neopterin	0.51 (240)	1.4 × 10^−3^	0.048(156)	n.s.	n.s.	X			
myo-inositol	0.49 (248)	1.7 × 10^−3^	0.48 (156)	6.4 × 10^−3^	n.s.	X	X		
gentisate	−0.5 (243)	1.8 × 10^−3^	−0.56 (154)	4.9 × 10^−4^	n.s.	X	X	X	
valine	0.49 (243)	1.9 × 10^−3^	0.15 (155)	n.s.	n.s.	X			
4-acetaminophen sulfate	0.58 (175)	2.5 × 10^−3^	0.36 (150)	0.034	n.s.	X	X		
arabitol/xylitol	−0.47 (248)	2.6 × 10^−3^	−0.15 (156)	n.s.	n.s.	X			

^1^ beta: effect size, N: number of samples with non-missing data, *p*-value from [3].^2^ Summary statistics from this study, using identical data preprocessing as in [3]. ^3^ *p*-value from the same model, but including eGFR as additional co-variate.

## Data Availability

All data used in this investigation are provided as Appendix A in Excel format.

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
