# Peer review of "Kidney Allograft Function Is a Confounder of Urine Metabolite Profiles in Kidney Allograft Recipients"

_metabolites, 2021, doi:10.3390/metabo11080533_

Round 1

Reviewer 1 Report

Major point

  1. Please revise your title to be specific.
  2. In acute rejection, the mechanisms of acute T-cell mediation rejection and antibody-mediated rejection are different, and it is known that the prognosis is worse in mixed rejection. Therefore, it is not reasonable to compare rejection and non-rejection by grouping ACR, AMR, and mixed type together. Since there are biopsy results in this study, please describe the relationship between the biopsy situation and metabolites.
  3. Please describe the relationship by presenting the relationship between eGFR and metabolites at the time of biopsy as well as changes in metabolites according to changes in eGFR.
  4. Please describe the relationship between the amount of proteinuria and metabolites.
  5. When acute rejection occurs, please describe the change of metabolites according to the treatment modalities.

Minor point

  1. In Table S1, the variable, “HLA-DSA3 at Transplant, N (%)” should be presented consistently.
  2. In Table S1, the number and percentage of mixed rejection biopsy group in the induction therapy were not match.
  3. In Table S2, the percentage of AMR biopsy group in the biopsy type was not match.
  4. In Table S2, the expression of the mean (SD) level of Banff Scores was not appropriate.

Reviewer 2 Report

The manuscript is very interesting and highlights important issues in the field of kidney rejection. The study design is appropriate and the results and conclusions are clearly presented. I recommend this manuscript for publication.

Reviewer 3 Report

Manuscript: Urine metabolomics of kidney graft rejection and the role of eGFR.

In the present study, authors evaluated metabolome of biopsy-matched urine cell-free supernatants of 192 samples from 153 kidney recipients using a new version of Metabolome platform enabling detection of more metabolites. The identification of urinary metabolites that can noninvasively discriminate rejection is of great importance. They used random forest method to pre-select 25 most discriminating metabolites between any rejection and no-rejection samples, followed by stability selection resulting in a model of 9 metabolites with AUC=0.918. They also found that many urine metabolites correlates with eGFR, especially from guanine pathway. They also tried to replicate the results of previous study (Suhre et al. JASN 2016) with 24 previously reported metabolites predictive of rejection and confirmed on their new patients cohort. They validated significance for 4 metabolites, while in the model with eGFR only 2 of them remained significant.

Major

  1. It is not clear if eGFR and CTOT4 signature are included in the model discriminating acute rejection from non-rejection. (Fig.6). If they were not selected by random forest method, then there is no reason to include them into model. You can make two models, one without eGFR and CTO4 signature and one including them to check how the performance of the model will change.
  2. Did you try to create a model discriminating ACR from AMR? And model discriminating PVAN from ACR? Of course, sample size is smaller ( 22 vs 16) and 22 vs 36). However, the ability to distinguish these diagnoses would be very helpful.
  3. In Fig.3 Volcano plots, you colored all significant metabolites regardless of effect size. I suppose that when effect size is small, then the metabolite contribution would be small.

Minor

  1. The first paragraph of the results belongs to Methods section, study cohort description. In addition, it should be mention that it was single center prospective ? study and also provide time period for patient enrolling.
  2. There are many spelling errors:

For example: p.10 l.346 weurine

p.10.l.349 pathologistsa…schrema…metabilita

p.10. l.363 Metablomics

  1. Supplementary Text 1 was not provided.
  2. In Fig.4 –correlation coef plus p values should be added.

Reviewer 4 Report

This is a nice paper. However, I have some comments.
The findings from this paper are excellent and worthy to review.
This manuscript contained some questions described below.
I think this paper is interesting, this review contributes to future's clinical

medicine largely.

I have some questions from a point of view of clinical medicine. Please tell us how to connect this data with daily clinical practice. I also think that interstitial tubular disorders are one of the important factors that determine renal prognosis. Please tell us about the relationship with markers in daily clinical practice, especially with urinary L-FABP.

Round 2

Reviewer 1 Report

I think that the authors have adequately corrected their paper according to the reviewer’s recommendations overall. I agree with the limitations of the revision from the authors.